# Understanding Neural Tangent Kernel Dynamics Through Its Trace Evolution

## Abstract

The Neural Tangent Kernel (NTK) has emerged as a valuable tool for analyzing the training and generalization properties of neural networks. While the behavior of the NTK in the infinite-width limit is well understood, a comprehensive investigation is still required to comprehend its dynamics during training in the finite-width regime. In this paper, we present a detailed exploration of the NTK's behavior through the examination of its trace during training.

By conducting experiments on standard supervised classification tasks, we observe that the NTK trace typically exhibits an increasing trend and stabilizes when the network achieves its highest accuracy on the training data. Additionally, we investigate the phenomenon of "grokking", which has recently garnered attention, as it involves an intriguing scenario where the test accuracy suddenly improves long after the training accuracy plateaus. To shed light on this phenomenon, we employ the NTK trace to monitor the training dynamics during grokking. Furthermore, we utilize the NTK trace to gain insights into the training dynamics of semi-supervised learning approaches, including the employment of exponential moving average mechanisms. Through these investigations, we demonstrate that the NTK, particularly its trace, remains a powerful and valuable tool for comprehending the training dynamics of modern finite-width neural networks.

## 1 Introduction

Deep neural networks have proven to be highly successful in a wide range of machine learning tasks, particularly in the domain of image classification. Understanding the dynamics of these networks during training and their generalization capabilities is of paramount importance to further advance their performance and interpretability. In recent years, the field of deep learning has seen the rise of a crucial analytical tool known as the Neural Tangent Kernel (NTK) (Jacot et al., 2018), primarily in the infinite-width limit under suitable initialization scales and infinitesimal learning rate. Empirically, researchers adopt neural networks of finite-width, therefore the behavior of the NTK in the finite-width regime necessitates a more comprehensive investigation.

Notably, in the infinite width limit, Jacot et al. (2018) show that when the network depth is fixed the NTK is determined by the initialization and is fixed throughout the whole training process. Theoretically, Hanin & Nica (2019); Littwin et al. (2021) show that the NTK at initialization is also influenced by the network depth even in the infinite width limit. Recently, Fort et al. (2020); Loo et al. (2022) empirically study the NTK of finite width neural network during the whole training process. By defining a "kernel distance" function, they show that after a rather quick chaotic data-dependent kernel learning process, the network learns a kernel that has the linearized training accuracy that matches the performance of the neural network. However, the defined "kernel distance" $S(K_1, K_2)$ (Fort et al., 2020; Loo et al., 2022) is not a distance function in mathematics as $S(K_1, K_2) = 0$ does not imply $K_1 = K_2$. But interestingly, one can show that if in addition $\text{Tr}(K_1) = \text{Tr}(K_2)$ then $K_1 = K_2$. Moreover, the NTK trace has an efficient approximation method, making it a feasible and orthogonal approach to prior works (Fort et al., 2020; Loo et al., 2022).

To understand the behavior of the NTK in finite-width neural networks, we conduct experiments that evaluate the NTK trace along the training process on 3 settings: standard supervised classification tasks, grokking, and semi-supervised learning. The latter 2 settings grokking and semi-supervised learning are closely related to standard supervised classification, which we will briefly discuss as

follows. Grokking is first observed in (Power et al., 2022), where a *supervised* classification task is trained to perform arithmetic tasks. It refers to a sudden and unexpected great improvement in test accuracy long after the training accuracy has reached a plateau. Semi-supervised learning uses a small amount of labeled data and a large amount of unlabeled data in training. It usually adopts the pseudo-labeling strategy, where during training the network will assign pseudo labels to some unlabeled data and thus transform it into a *supervised* learning problem.

Our contributions can be summarized as follows:

1. We present an efficient approximation for NTK trace and link it to the margin of a kernel SVM problem.

2. On standard supervised classification tasks, we find the NTK trace usually rises and stabilizes when top training accuracy is met. We also observe that in the grokking settings, when the NTK trace is stabilized the test accuracy attains its maximum.

3. The NTK trace also helps us understand training dynamics in semi-supervised learning approaches, including the use of the exponential moving average (EMA) mechanism.

## 2 RELATED WORK

**Neural Tangent Kernel.**   Jacot et al. (2018) provide neural tangent kernel (NTK) as a tool to understand the dynamics of neural networks training under suitable initialization and small learning rates by considering the infinite-width limit. In this limiting regime, the NTK becomes deterministic and fixed during training when the width is taken to infinity (Zou & Gu, 2019; Ji & Telgarsky, 2019; Chen et al., 2019). As modern neural networks are in the finite-width regime, researchers have gained more and more interest in the behavior of NTK in modern finite-width architectures (Shan & Bordelon, 2021; Atanasov et al., 2021; Seleznova et al., 2024; Hanin & Nica, 2019; Littwin et al., 2021; Fort et al., 2020; Loo et al., 2022). For example, Hanin & Nica (2019); Littwin et al. (2021) have given finite-width (and depth) correction of neural network's NTK at initialization. And Fort et al. (2020); Loo et al. (2022) study the evolution of NTK during modern finite-width neural network training using a "kernel distance". In this paper, we study dynamics of NTK using its trace.

**Neural network training dynamics.**   Researchers have found that there has usually been some implicit optimization process during the supervised training dynamics of neural networks (Nacson et al., 2019; Xu et al., 2021; Soudry et al., 2018; Banburski et al., 2019; Lyu & Li, 2019; Blanc et al., 2020). For example, Lyu & Li (2019) show the network implicitly maximizes the margin during the training using sgd under loss like cross-entropy loss and Blanc et al. (2020) study the implicit regularization of the network under MSE loss using sgd. Very recently, Power et al. (2022) find cases in supervised learning that the network generalizes long after it overfits the training data, which they termed the phenomenon grokking. There have been attempts to understand the training dynamics of grokking: Liu et al. (2022a) study the dynamics using physics-inspired effective theory, Nanda et al. (2023) study the process using trigonometric series, and Tan & Huang (2023) study the process using the network's robustness. In this paper, we study the dynamics of neural networks and their associated NTK using the trace of NTK.

## 3 PRELIMINARY

We will first introduce some basic knowledge to better understand NTK. Suppose the training dataset as $\mathcal{X} = \{(x_i, y_i)\}_{i=1}^N$ with scalar labels $\{y_i\}_{i=1,\dots,N} \subset \mathbb{R}$ and a loss function $l : \mathbb{R} \times \mathbb{R} \to \mathbb{R}$. Then the empirical training loss, defined on functions $f : \mathbb{R}^{n_1} \to \mathbb{R}$, is given by

$$l(f) = \sum_{i=1}^N l\left(f\left(x_i\right), y_i\right).$$

When the network $f(\cdot; w) : \mathbb{R}^{n_1} \to \mathbb{R}$ is trained by minimizing the loss $l(f)$ via gradient flow, then the network parameters $(w(t))_{t \geq 0}$ will change according to the following ordinary differential equation:

$$\partial_t w(t) = -\nabla_w l(f(\cdot; w(t))). \tag{1}$$

Therefore during training, the output function of the neural network for any $x \in \mathbb{R}^{n_1}$ follows another differential equation given in terms of the Neural Tangent Kernel (NTK) (Lee et al., 2019):

$$\partial_t f(x; w(t)) = - \sum_{i=1}^{N} K(x, x_i; w) \, \partial_z l(z, y_i) \Bigg|_{z = f(x_i; w(t))},$$

where $K(\cdot, \cdot; w)$ is the Neural Tangent Kernel defined as follows:

$$K(x, y; w) = \sum_{k=1}^{d} \partial_{w_k} f(x; w) \partial_{w_k} f(y; w) = \langle \nabla f(x; w), \nabla f(y; w) \rangle.$$

The above equation shows that the NTK determines the training dynamics of $f(\cdot; w(t))$ in the function spaces $\mathbb{R}^{n_1} \to \mathbb{R}$ during gradient flow training.

For a neural network $f(x; w) \in \mathbb{R}^K$ with vector outputs (logits) with parameter $w \in \mathbb{R}^d$, we will define the NTK as follows. Its Jacobian matrix can be computed as $J(x, w) \in \mathbb{R}^{K \times d}$. Suppose the dataset as $\mathcal{X} = \{(x_i, y_i)\}_{i=1}^{N}$ and denote $J(\mathcal{X}, w) = [J^\top(x_1, w), \cdots, J^\top(x_N, w)]^\top \in \mathbb{R}^{NK \times d}$. Then the empirical neural tangent kernel on $\mathcal{X}$ can be computed as $K(\mathcal{X}, \mathcal{X}; w) = J(\mathcal{X}, w) J^\top(\mathcal{X}, w)$. All the proofs of this paper's theorems can be found in Appendix A.

# 4 EVOLUTION OF NTK (TRACE)

We are interested in evaluating the dynamics of NTK during network training, as the NTK is a matrix, we intend to evaluate its trace (a scalar) $\frac{\mathrm{Tr}(K(\mathcal{X}, \mathcal{X}; w))}{|\mathcal{X}| K}$ during training to reflects its dynamic.

Jacot et al. (2018) show that NTK is constant along training in the infinite-width limit. As we are interested in the dynamics of finite-width NTK, we will abbreviate each time-step's $K(\mathcal{X}, \mathcal{X}; w(t))$ as $K_t(\mathcal{X}, \mathcal{X})$. Note that directly calculating the NTK matrix for the whole dataset is computational and memory inefficient, we will provide the following theorem 4.1 that can efficiently approximate the tendency of the trace of NTK.

**Theorem 4.1.** $\mathbb{E}_{\Delta \sim \epsilon \mathcal{N}(0, I_d)} \| f(\mathcal{X}; w + \Delta) - f(\mathcal{X}; w) \|_F^2 \sim \epsilon^2 \, \mathrm{Tr}(K(\mathcal{X}, \mathcal{X}; w)).$

We can immediately obtain an approximate calculation of $\mathrm{Tr}(K(\mathcal{X}, \mathcal{X}; w))$ as $\mathrm{Tr}(K(\mathcal{X}, \mathcal{X}; w)) \approx \frac{1}{\epsilon^2} \| f(\mathcal{X}; w + \Delta) - f(\mathcal{X}; w) \|_F^2$, where $\Delta \sim \epsilon \mathcal{N}(0, I_d)$. We plot the evolution of NTK traces on 3 widely adopted benchmarks in Figure 1 with the backbone ResNet18 and sgd with momentum. More training details and effect of different learning rate, momentum, and initialization can be found in Appendix B. We can see that the trend of NTK trace is similar among datasets as the trace continues to increase until the training accuracy reaches its maximum. This shows that the NTK matrix stabilizes until training reaches the optimal point.

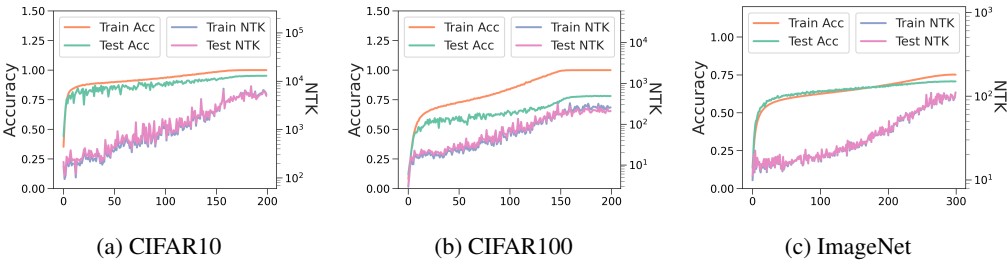

(a) CIFAR10        (b) CIFAR100        (c) ImageNet

Figure 1: Evolution of NTK trace on different datasets using ResNet18.

In this paper, we mainly consider using cross-entropy loss to train a classification problem, it may have different outcomes when facing other settings. When given a dataset $\mathcal{X} = \{(x_i, y_i)\}_{i=1}^{N}$, the loss is $\frac{1}{N} \sum_{i=1}^{N} \mathrm{H}(\mathrm{onehot}(y_i), \mathrm{prob}(x_i))$, where we use $\mathrm{H}(p, q) = -p \log q$ to denote the cross-entropy loss between two probability distributions $p$ and $q$, and the generated probability of a sample $x$ under

model $f(x; w)$ is defined as $\text{prob}(x; w) == [\frac{e^{f_1(x;w)}}{\sum_{j=1}^{K} e^{f_j(x;w)}}, \cdots, \frac{e^{f_K(x;w)}}{\sum_{j=1}^{K} e^{f_j(x;w)}}]^\top$, which we usually abbreviate it as $\text{prob}(x)$.

Another question remains, what property of the neural network does the NTK trace value $\text{Tr}(K_t(\mathcal{X}, \mathcal{X})))$ reflect during training? We will use the setting in (Lyu & Li, 2019) to gain some insights into the above question. We will summarize the key technical assumptions in (Lyu & Li, 2019) that will be useful in this paper, for more detailed discussions of other assumptions, please refer to the initial paper (Lyu & Li, 2019).

1. The problem is a binary classification problem, i.e. $y_i \in \{+1, -1\}$.
2. For each sample $(x_i, y_i)$, the loss is the logistic loss (binary cross-entropy loss), i.e. $l((x_i, y_i)) = \log(1 + e^{-y_i f(x_i; w)})$.

Then we have the following theorem which connects the dynamics of the neural network to the solution of an NTK kernel SVM.

**Theorem 4.2** ((Lyu & Li, 2019)). *Under some technical assumptions and assuming the network is trained by gradient flow, then any limit point $\bar{w}$ of $\{\frac{w(t)}{\|w(t)\|}|t > 0\}$ is along the max-margin direction for a hard-margin SVM with kernel $K(x, y) = \langle \nabla f(x; \bar{w}), \nabla f(y; \bar{w}) \rangle$.*

*This means that for some $\beta > 0$, $\beta\bar{w}$ is the optimal solution to the following optimization problem:*

$$\min \quad \frac{1}{2}\|w\|^2$$
$$s.t. \quad y_i\langle w, \nabla f(x_i; \bar{w}) \rangle \geq 1 \quad i = 1, 2, \cdots, N \tag{2}$$

Considering the dual of optimization problem (2), we will obtain an optimization problem as follows:

$$\max \quad \sum_{i=1}^{N} \alpha_i - \frac{1}{2}\sum_{i,j} \alpha_i \alpha_j y_i y_j K(x_i, x_j; \bar{w})$$
$$s.t. \quad \alpha \geq 0 \tag{3}$$

Motivated by (Seleznova et al., 2024), we will make the following simplification assumptions:

1. If $y_i = y_j$, then $K(x_i, x_j; \bar{w}) = x$.
2. If $y_i \neq y_j$, then $K(x_i, x_j; \bar{w}) = 0$.

Collect the dual variables in $\alpha$ which corresponds to $y_i = +1$ as $\alpha^1$ and $y_i = -1$ as $\alpha^2$. The optimization problem (3) simplifies as follows:

$$\max \quad -\frac{x}{2}(\sum_i \alpha_i^1)^2 + (\sum_i \alpha_i^1) + -\frac{x}{2}(\sum_i \alpha_i^2)^2 + (\sum_i \alpha_i^2)$$
$$s.t. \quad \alpha^1, \alpha^2 \geq 0 \tag{4}$$

It is easy to see the optimal objective value for the optimization problem (4) is $\frac{1}{x}$. By noticing that Under the assumption "If $y_i = y_j$, then $K(x_i, x_j; \bar{w}) = x$", we can deduce that $Nx = \text{Tr}(K(\mathcal{X}, \mathcal{X}; \bar{w}))$. Therefore, we find that $1/\frac{\text{Tr}(K(\mathcal{X}, \mathcal{X}; \bar{w}))}{N}$ is a good approximate optimal objective value for (2) by utilizing the duality of (2) and (3). Therefore the results in Figure 1 show that the network is implicitly maximizing the margin of kernel SVM during training.

Notably, (Fort et al., 2020; Loo et al., 2022) also study the dynamics of NTK in the finite-width neural network through the following "kernel distance" between two kernel matrices: $S(K_1, K_2) = 1 - \frac{\text{Tr}(K_1^\top K_2)}{\|K_1\|_F \|K_2\|_F}$.

(Fort et al., 2020; Loo et al., 2022) define the kernel velocity as $v(t) = \frac{S(K(\mathcal{X}, \mathcal{X}; w(t)), K(\mathcal{X}, \mathcal{X}; w(t+dt)))}{dt}$.

If the total training time is $T_1$, (Fort et al., 2020; Loo et al., 2022) find the following facts:

1. There exists a time $T_0$, where $v(t) = 0$ ($\forall t > T_0$).
2. $S(K(\mathcal{X}, \mathcal{X}; w(T_0)), K(\mathcal{X}, \mathcal{X}; w(T_1))) = 0$.

From the above facts, one may find that $S(K_{t+1}(\mathcal{X}, \mathcal{X}), K_t(\mathcal{X}, \mathcal{X})) = 0$ ($t \geq T_0$). However, the defined "kernel distance" is not a distance function in mathematics as $S(K_1, K_2) = 0$ does not imply $K_1 = K_2$. Note we are interested in the dynamics of $K_t(\mathcal{X}, \mathcal{X})$, the following theorem 4.3 shows that $\text{Tr}(K_t(\mathcal{X}, \mathcal{X}))$ together with the "kernel distance" can characterize whether the NTK stabilize or not. Figure 2 together with Figure 1 further validates that the "kernel distance" and NTK trace give complementary characterizations of the dynamics of NTK.

**Theorem 4.3.** *Suppose* $S(K_{t+1}(\mathcal{X}, \mathcal{X}), K_t(\mathcal{X}, \mathcal{X})) = 0$ ($t \geq T_0$). *Suppose* $T > T_0$, *then* $K_i(\mathcal{X}, \mathcal{X}) = K_j(\mathcal{X}, \mathcal{X})$ ($i, j \geq T$) *iff* $\text{Tr}(K_i(\mathcal{X}, \mathcal{X})) = \text{Tr}(K_j(\mathcal{X}, \mathcal{X}))$ ($i, j \geq T$).

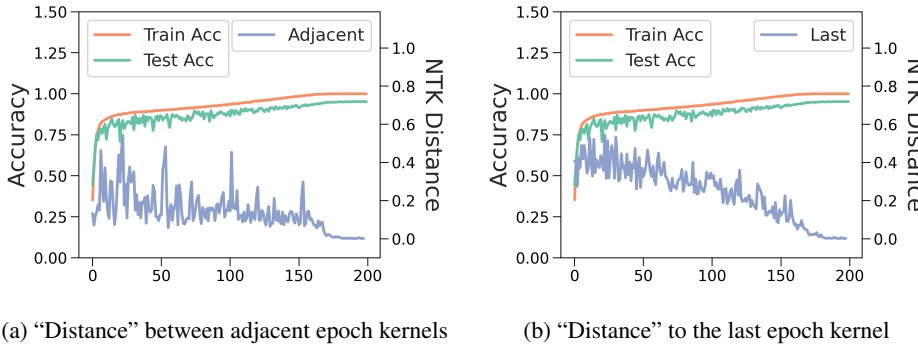

(a) "Distance" between adjacent epoch kernels      (b) "Distance" to the last epoch kernel

Figure 2: Accuracy and NTK "kernel distance".

## 5 UNDERSTANDING DYNAMICS OF OTHER TRAINING SCENARIOS USING NTK TRACE

### 5.1 LAYERWISE DYNAMICS DURING TRAINING

After investigating the behavior of the NTK trace in the previous section, we now focus on understanding the behavior of inner layers in the neural network. To do this, we depict the architecture of the network in Figure 3. We specifically examine the outputs of three layers: the front layer (Layer 1 in Figure 3), which is close to the input picture, the middle layer (Layer 3 in Figure 3), and the final layer, whose output is the logit. For simplicity, we denote these layers as $f_1$, $f_2$, and $f_3$, respectively.

In addition to calculating the NTK trace for the logits, as described in theorem 4.1, we also compute $\frac{\mathbb{E}_{\Delta \sim \epsilon \mathcal{N}(0, I_d)} |f(\mathcal{X}; w+\Delta) - f(\mathcal{X}; w)|_F^2}{\epsilon^2 |\mathcal{X}| K_i}$ ($i = 1, 2$), where $K_i$ is the dimension of the layer output. We present the results in Figure 4.

From Figure 4, we observe that each layer exhibits a distinct evaluation pattern. The last layer stabilizes the latest, while the middle layer stabilizes the fastest. This discrepancy suggests that different layers play different roles in the neural network. As the last layer stabilizes slowly, it is crucial to track the NTK trace of the logits to gain a comprehensive understanding of the entire training dynamics.

### 5.2 THE DYNAMICS OF GROKKING

In this section, we consider a special setting in supervised classification, called grokking (Liu et al., 2022b; Nanda et al., 2023). It refers to a strange phenomenon that the test accuracy suddenly increases long after training accuracy reaches $100\%$, Figures 5 and 6 depict two scenarios for this phenomenon. To conduct our experiments, we consider two canonical grokking settings. The first setting focuses on the MNIST image classification task (Liu et al., 2022b). For this task, we adopt a ReLU MLP architecture with a width of 200 and a depth of 3. The loss function used is mean squared error (MSE) loss. To optimize the network, we employ the AdamW optimizer with a learning rate of

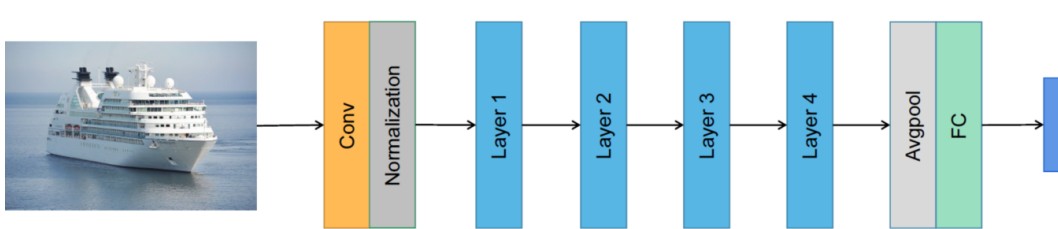

Figure 3: The architecture of ResNet18.

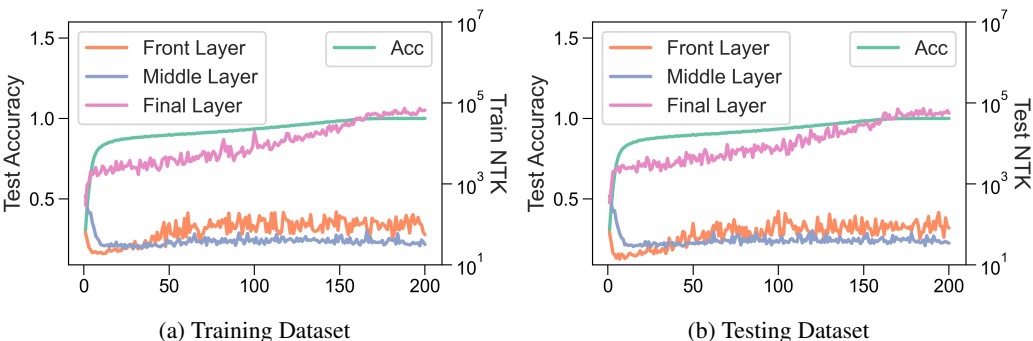

(a) Training Dataset                    (b) Testing Dataset

Figure 4: The evolution of NTK trace among different layers, on both training and testing datasets.

0.001, and we use a batch size of 200 during the training process. In the second setting, we explore the modulo addition dataset (Nanda et al., 2023). Here, we train transformers to perform addition modulo $P = 113$. The input format follows the pattern of $a + b =$, where $a$ and $b$ are encoded as $P$-dimensional one-hot vectors. The output $c$ is extracted from the special token $=$. Our model for this task consists of a one-layer ReLU transformer. The token embeddings have hidden dimension $d = 128$, and we incorporate learned positional embeddings. The model comprises 4 attention heads with a dimension of $\frac{d}{4} = 32$, and an MLP with 512 hidden units. We use a training dataset containing 30% of all possible input pairs, and the loss function employed is cross-entropy loss.

In our analysis, we observe that the estimated NTK trace, given by $\frac{1}{\epsilon^2}|f(\mathcal{X}; w + \Delta) - f(\mathcal{X}; w)|_F^2$, bears a resemblance to the robustness metric (Tan & Huang, 2023). Specifically, we can define the logit perturb distance on the training data as $\frac{1}{\epsilon^2}|f(\mathcal{X} + \hat{\Delta}; w) - f(\mathcal{X}; w)|_F^2$, where $\hat{\Delta}$ represents a Gaussian perturbation applied to the input data. We consider both the NTK trace and the logit perturb distance and present them in Figures 5 and 6.

Our analysis reveals that the NTK traces on both the train and test datasets stabilize only when the test accuracy approaches its maximum value. This finding aligns with the observations made in standard supervised classification, as discussed in section 4, indicating that the underlying dynamics of the network still changes when the training accuracy reaches 100% and stabilizes only after the test accuracy saturates. Note the experiments in Figure 5 utilizes cross-entropy loss and the NTK trace continuous to increase before the test accuracy first reaches 100%, making it similar to the observation in section 4. Interestingly, we also observe that the trend of the NTK trace and the logit perturb distance exhibits several similarities, which could be attributed to the similarities in their respective defining formulas.

### 5.3 SEMI-SUPERVISED LEARNING DYNAMICS

We will first present a simple review of a canonical semi-supervised learning method FixMatch. Given a batch of $B$ labelled samples $\mathcal{X}_l = \{(x_i^1, y_i^1)\}_{i=1}^B$, there will also be $\mu B$ unlabelled samples

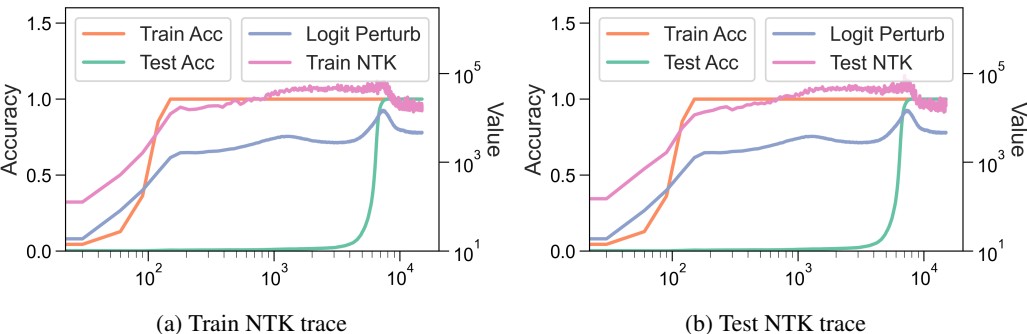

Figure 5: The evolution of NTK trace and logit perturbation distance on Modular Addition dataset.

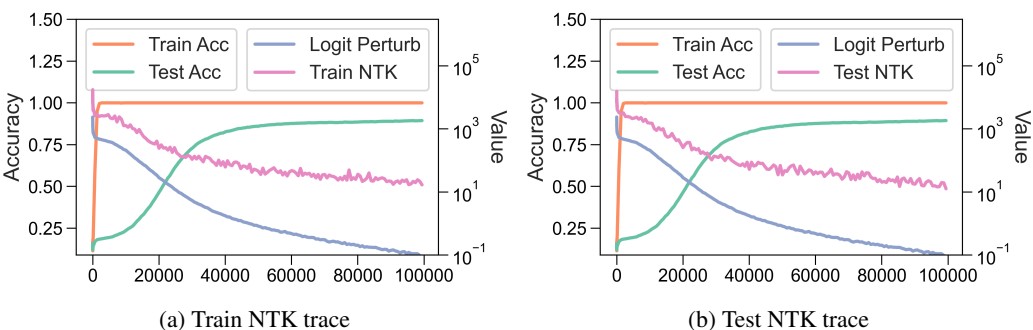

Figure 6: The evolution of NTK trace and logit perturbation distance on MNIST dataset.

$\mathcal{X}_u = \{(x_i^2)\}_{i=1}^{\mu B}$ involved in the training. FixMatch loss utilizes two types of augmentations: weak and strong, defined by $\omega(\cdot)$ and $\Omega(\cdot)$. The loss consists of two parts: the supervised loss $l_1$ and the unsupervised loss $l_2$.

$$l_1 = \frac{1}{B} \sum_{i=1}^{B} \mathrm{H}(\mathrm{onehot}(y_i^1), \mathrm{prob}(\omega(x_i^1))). \tag{5}$$

$$l_2 = \frac{1}{\mu B} \sum_{i=1}^{\mu B} \mathbb{I}(\max(\mathrm{prob}(\omega(x_i^2))) \geq \phi)\, \mathrm{H}(\mathrm{onehot}(\hat{y}_i^2), \mathrm{prob}(\Omega(x_i^2))), \tag{6}$$

where $\hat{y}_i^2 = \arg\max \mathrm{prob}(\omega(x_i^2))$.

### 5.3.1 EVALUATING THE DYNAMICS ON LABELED AND UNLABELLED DATA

Semi-supervised learning involves both labeled and unlabeled samples, the key difference is that compared to the supervised case where the model is only trained on labeled samples, the training

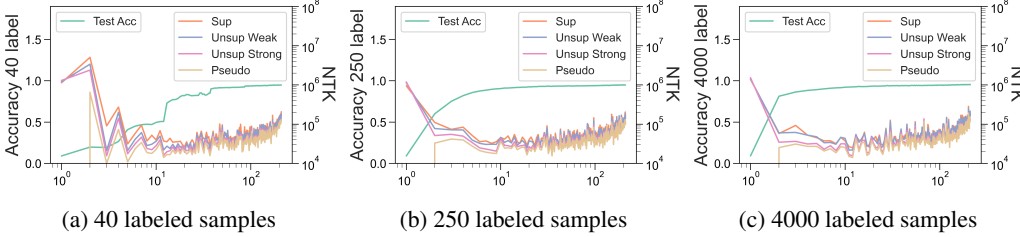

Figure 7: The evolution of NTK trace estimations of (weakly-augmented) labeled, (weakly and strongly augmented) unlabeled, and pseudo-labeled samples during training. We consider three configurations with different numbers of labeled samples.

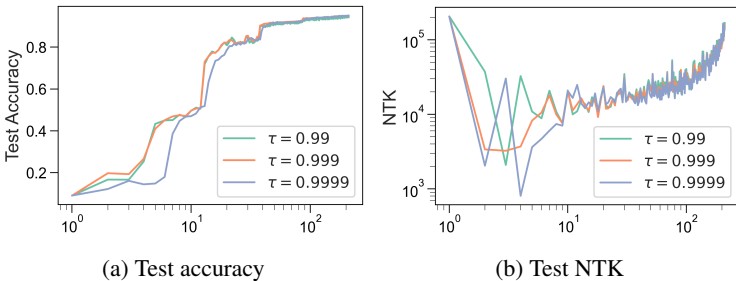

(a) Test accuracy  (b) Test NTK

Figure 8: The evolution of test accuracy and NTK trace under different EMA coefficient $\tau$.

process of semi-supervised learning is much more complex. Specifically, it involves both the (weakly augmented-)labeled and strongly augmented (unlabeled-)samples that are pseudo-labeled in the training. Motivated by the discussions in the supervised learning settings, we are interested in the dynamics of NTK trace of the following samples: 1) All the weakly augmented-labeled samples. 2) All The weakly augmented-unlabeled samples. 3) All the strongly augmented-unlabeled samples. 4) All the strongly augmented-unlabeled samples that are pseudo-labeled. Denote $\hat{\mathcal{X}}_u$ as the set of samples that are pseudo-labeled, i.e. those $x_i^2$ which satisfies $\max(\text{prob}(\omega(x_i^2))) \geq \tau$. Then we are interested in the following 4 quantities: 1) Supervised (labeled) samples (Sup) $\frac{\text{Tr}(K_t(\omega(\mathcal{X}_l),\omega(\mathcal{X}_l)))}{BK}$. 2) Unsupervised (unlabeled) weakly augmented-samples (Unsup Weak) $\frac{\text{Tr}(K_t(\omega(\mathcal{X}_u),\omega(\mathcal{X}_u)))}{\mu BK}$. 3) Unsupervised (unlabeled) strongly augmented-samples (Unsup Strong) $\frac{\text{Tr}(K_t(\Omega(\mathcal{X}_u),\Omega(\mathcal{X}_u)))}{\mu BK}$. 4) Pseudo-labeled samples (Pseudo) $\frac{\text{Tr}(K_t(\Omega(\hat{\mathcal{X}}_u),\Omega(\hat{\mathcal{X}}_u)))}{\mu BK}$.

In our experimental setup, we have closely followed the configuration used in FixMatch (Sohn et al., 2020). Our approach involves the utilization of a sgd optimizer with a momentum of 0.9. Additionally, we have employed a cosine learning rate scheduler, initialized at 0.03, to dynamically adjust the learning rate during training. The total number of training steps is set to $2^{20}$, with evaluations conducted at every 5000 steps and makes the total number of evaluation steps to be 210. For the labeled samples, we have chosen a batch size of 64, while maintaining a ratio $\mu$ of 7 between the unlabeled and labeled samples. Furthermore, we have set the threshold value $\phi$ to 0.95 to guide the decision-making process. The weak and strong augmentation functions are used following the RandAugment approach (Cubuk et al., 2020). Lastly, we have employed the WideResNet-28-2 architecture as the underlying backbone model for our experimental investigations.

Figure 7 demonstrates the NTK trace for varying label quantities, with a range of 40, 250, and 4000 labeled samples in total. Notably, the changing patterns of each quantity exhibit remarkable similarities across the different label quantities. To enhance clarity, a logarithmic scale is employed for the evaluation steps. The figure reveals that the accuracy undergoes slow changes after approximately 20 steps, which corresponds to around 10% of the total training steps. Simultaneously, all NTK traces experience a significant decrease, with distinct values for each of the four traces. Subsequently, throughout the remaining training process, all NTK traces exhibit an increasing trend and converge towards similar values. This phenomenon can be attributed to the scarcity of labeled data in the early stages of training, which leads to a less stable training process. Furthermore, it indicates that the weakly augmented labeled and unlabeled data share similar NTK traces, implying that the predicted pseudo-labels may have similar distributions to the labeled ones. Additionally, the NTK traces of weakly and strongly augmented samples are similar, suggesting that the network generalizes well on strongly augmented samples. Finally, the NTK traces of strongly augmented samples and pseudo-labeled samples exhibit resemblances, indicating a high efficiency of pseudo-labeling.

### 5.3.2 INVESTIGATING THE DYNAMICS OF EXPONENTIAL MOVING AVERAGE MECHANISM

To enhance the prediction accuracy, in semi-supervised learning an exponential moving average (EMA) technique is usually employed. Specifically, at each step $n$, denote the EMA model as $z_n^2$ and the initial model as $z_n^1$, the update rule for the EMA model with momentum $\tau$ is $z_{n+1}^2 = \tau z_{n+1}^2 + (1-\tau)z_n^1$.

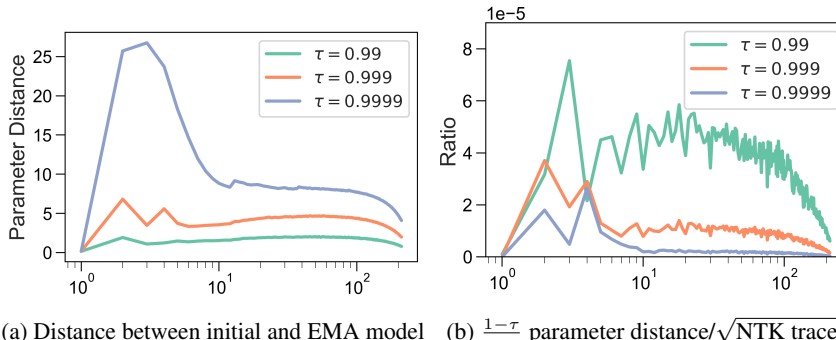

(a) Distance between initial and EMA model     (b) $\frac{1-\tau}{\tau}$ parameter distance/$\sqrt{\text{NTK trace}}$

Figure 9: Parameter distance and NTK ratio.

We plot the accuracy and NTK trace of EMA model $z_n^2$ on the test dataset under different $\tau$ in Figure 8 using FixMatch on CIFAR-10 with 40 labels. The best accuracy for different $\tau$ is listed as: $\tau = 0.99$, accuracy$= 94.73$. $\tau = 0.999$, accuracy$= 94.97$. $\tau = 0.9999$, accuracy$= 95.13$. The evolution of test NTK is similar to the observation in section 5.3.1 with a first chaotic phase and quickly turns into an increasing phase.

For the behavior of accuracy under different momentum parameters $\tau$ in EMA, we will present an intuitive theoretical understanding of the phenomenon as follows using the gradient flow.

**Theorem 5.1.** *Denote the parameter for the initial model along training as $z^1(t)$ and the parameter for the EMA updated model with momentum $\tau$ as $z_\tau^2(t)$. Assume $z_\tau^2(t)$ also satisfies the gradient flow equation (1) and the training loss is a cross-entropy loss for input pair $(x_i, y_i)$, where $y_i$ is a ground-truth label or pseudo-label. Then we have the following bound on the average error:*

$$\frac{1-\tau}{\tau}\|z^1(t) - z_\tau^2(t)\| \le \sqrt{\frac{\text{Tr}(K(\mathcal{X}, \mathcal{X}; z_\tau^2(t)))}{N}}\sqrt{\frac{\sum_{i=1}^{N}\|onehot(y_i) - prob(x_i; z_\tau^2(t))\|^2}{N}}. \quad (7)$$

We therefore plot the $l_2$ distance between initial model and EMA model parameters $\|z^1(t) - z_\tau^2(t)\|$ in Figure 9 (a) and the ratio $\frac{1-\tau}{\tau}\|z^1(t) - z_\tau^2(t)\|/\sqrt{\frac{\text{Tr}(K(\mathcal{X}, \mathcal{X}; z_\tau^2(t)))}{N}}$ in Figure 9 (b). Recall that the test accuracy corresponds to $\tau = 0.99, 0.999, 0.9999$ is $94.73, 94.97, 95.13$. From Figure 9 (b), we can see the ratio decreases after the initial unstable training phase. The ratio has a decreasing order with the size of $\tau$ and is getting close at the end of training, this matches the fact that test accuracy is close but with decreasing order with the size of $\tau$. These observations show that the inequality (7) offers valuable insight into the dynamic of the EMA model.

## 6 CONCLUSION

Through the analysis of the NTK trace, we gained valuable insights into the network's training progress and its correlation with accuracy. Our findings in standard supervised image classification settings demonstrated that the NTK trace typically exhibited an increasing trend, eventually stabilizing when the network achieved its highest accuracy on the training data. This observation underscores the intrinsic connection between the NTK dynamics and the network's training performance. Furthermore, our exploration of the phenomenon referred to as "grokking" yielded further insights. Through the utilization of the NTK trace, we closely monitored the training dynamics in grokking scenarios and made a noteworthy observation: the test accuracy reaches its peak when the NTK trace stabilizes. In the final phase of our study, we investigated the training dynamics of semi-supervised learning, with a particular emphasis on examining the effectiveness of exponential moving average (EMA) mechanisms. By leveraging the NTK trace, we gained a deeper and more accurate understanding of the intricate behavior displayed in semi-supervised learning scenarios, as well as the influence of EMA on the training process. This analysis provided valuable insights into the dynamics of semi-supervised learning and shed light on the role of EMA in enhancing training performance. One limitation of our work is not analyzing the whole property of NTK, just its trace.

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

## A  PROOFS OF THEOREMS

**Theorem A.1.** $\mathbb{E}_{\Delta \sim \epsilon \mathcal{N}(0, I_d)} \| f(\mathcal{X}; w + \Delta) - f(\mathcal{X}; w) \|_F^2 \sim \epsilon^2 \operatorname{Tr}(K(\mathcal{X}, \mathcal{X}; w))$.

*Proof.* From Taylor expansion,

$\mathbb{E}_{\Delta \sim \epsilon \mathcal{N}(0, I_d)} \| f(\mathcal{X}; w + \Delta) - f(\mathcal{X}; w) \|_F^2 = \sum_{i=1}^{N} \mathbb{E}_{\Delta \sim \epsilon \mathcal{N}(0, I_d)} \| f(x_i; w + \Delta) - f(x_i; w) \|_F^2 \sim$
$\sum_{i=1}^{N} \mathbb{E}_{\Delta \sim \epsilon \mathcal{N}(0, I_d)} \| J(x_i, w) \Delta \|_F^2 = \sum_{i=1}^{N} \mathbb{E}_{\Delta \sim \epsilon \mathcal{N}(0, I_d)} \operatorname{Tr}(J(x_i, w) \Delta \Delta^\top J^\top(x_i, w)) =$
$\sum_{i=1}^{N} \operatorname{Tr}(\mathbb{E}_{\Delta \sim \epsilon \mathcal{N}(0, I_d)} J(x_i, w) \Delta \Delta^\top J^\top(x_i, w)) = \epsilon^2 \sum_{i=1}^{N} \operatorname{Tr}(J(x_i, w) J^\top(x_i, w)) =$
$\epsilon^2 \operatorname{Tr}(J(\mathcal{X}, w) J^\top(\mathcal{X}, w)) = \epsilon^2 \operatorname{Tr}(K(\mathcal{X}, \mathcal{X}; w))$. $\qquad\square$

**Theorem A.2.** *Suppose* $S(K_{t+1}(\mathcal{X}, \mathcal{X}), K_t(\mathcal{X}, \mathcal{X}))) = 0$ $(t \geq T_0)$. *Suppose* $T > T_0$, *then* $K_i(\mathcal{X}, \mathcal{X}) = K_j(\mathcal{X}, \mathcal{X})$ $(i, j \geq T)$ *iff* $\operatorname{Tr}(K_i(\mathcal{X}, \mathcal{X})) = \operatorname{Tr}(K_j(\mathcal{X}, \mathcal{X}))$ $(i, j \geq T)$.

*Proof.* Define $a_t = \operatorname{vec}(K_t(\mathcal{X}, \mathcal{X}))$. Denote $M_j = \operatorname{Tr}(K_j(\mathcal{X}, \mathcal{X}))$ and $B = \frac{K_{t_0}(\mathcal{X}, \mathcal{X})}{\| K_{t_0}(\mathcal{X}, \mathcal{X}) \|_F}$. From the assumption, we know that $\frac{a_t}{\| a_t \|_2} = \frac{a_{t_0}}{\| a_{t_0} \|_2}$ $(t \geq t_0)$. Thus $\frac{K_j(\mathcal{X}, \mathcal{X})_{mm}}{\| K_j(\mathcal{X}, \mathcal{X}) \|_F} = B_{mm}$. Then $M_j = \sum_m K_j(\mathcal{X}, \mathcal{X})_{mm} = \sum_m \| K_j(\mathcal{X}, \mathcal{X}) \|_F B_{mm} = \| K_j(\mathcal{X}, \mathcal{X}) \|_F \operatorname{Tr}(B)$. Then $K_j(\mathcal{X}, \mathcal{X})_{mn} = \| K_j(\mathcal{X}, \mathcal{X}) \|_F B_{mn} = \frac{M_j}{\operatorname{Tr}(B)} B_{mn}$. Thus $K_j(\mathcal{X}, \mathcal{X}) = \frac{M_j}{\operatorname{Tr}(B)} B$. Thus the conclusion follows. $\qquad\square$

**Theorem A.3.** *Denote the parameter for the initial model along training as* $z^1(t)$ *and the parameter for the EMA updated model with momentum* $\tau$ *as* $z_\tau^2(t)$. *Assume* $z_\tau^2(t)$ *also satisfies the gradient flow equation (1) and the training loss is a cross-entropy loss for input pair* $(x_i, y_i)$, *where* $y_i$ *is a ground-truth label or pseudo-label. Then we have the following bound*

$$\frac{1 - \tau}{\tau} \| z^1(t) - z_\tau^2(t) \| \leq \sqrt{\frac{\operatorname{Tr}(K(\mathcal{X}, \mathcal{X}; z_\tau^2(t)))}{N}} \sqrt{\frac{\sum_{i=1}^{N} \| onehot(y_i) - prob(x_i; z_\tau^2(t)) \|^2}{N}}. \quad (8)$$

*Proof.* The update formula for EMA model is as follows:

$$z_\tau^2(t + dt) = \tau z_\tau^2(t) + (1 - \tau) z^1(t).$$

Then we can obtain the following ordinary differential equation

$$\partial_t z_\tau^2(t) = \frac{1 - \tau}{\tau} (z^1(t) - z_\tau^2(t)). \quad (9)$$

Assume $z_\tau^2(t)$ satisfies equation (1), then equation (9) simplifies to the following

$$-\nabla_w l(f(\mathcal{X}; z_\tau^2(t))) = \frac{1 - \tau}{\tau} (z^1(t) - z_\tau^2(t)). \quad (10)$$

Using the expression of $l$, one may derive the following

$$\frac{1}{N} \sum_{i=1}^{N} J^\top(x_i, z_\tau^2(t))(onehot(y_i) - prob(x_i; z_\tau^2(t)) = \frac{1 - \tau}{\tau} (z^1(t) - z_\tau^2(t)), \quad (11)$$

where $prob(x_i; z_\tau^2(t)) = [\frac{e^{f_1(x_i; z_\tau^2(t))}}{\sum_{j=1}^{K} e^{f_j(x_i; z_\tau^2(t))}}, \cdots, \frac{e^{f_K(x_i; z_\tau^2(t))}}{\sum_{j=1}^{K} e^{f_j(x_i; z_\tau^2(t))}}]^\top$.

Then by the triangular inequality for norm and Cauchy–Schwarz inequality, we can obtain the following bound

$$\frac{1 - \tau}{\tau} \| z^1(t) - z_\tau^2(t) \| \leq \frac{1}{N} \sum_{i=1}^{N} \| J^\top(x_i, z_\tau^2(t)) \|_F \| onehot(y_i) - prob(x_i; z_\tau^2(t)) \|$$

$$\leq \sqrt{\frac{\operatorname{Tr}(K(\mathcal{X}, \mathcal{X}; z_\tau^2(t)))}{N}} \sqrt{\frac{\sum_{i=1}^{N} \| onehot(y_i) - prob(x_i; z_\tau^2(t)) \|^2}{N}}. \quad (12)$$

**Remark:** Similar arguments also hold for MSE loss. $\qquad\square$

# B    MORE EMPIRICAL RESULTS ON STANDARD SUPERVISED CLASSIFICATION

The default settings for the experiment are as follows: the dataset used is CIFAR10, and the model employed is ResNet18. The parameter $\epsilon$ for NTK trace calculation is set to 0.01. The learning rate is initialized to 0.1, and momentum is set to 0.9. Weight decay is set to $5 * 10^{-4}$. The batch size for training is set to 128, and the total number of training epochs is 200. Additionally, a cosine learning rate decay strategy is employed during training. For CIFAR100 dataset, the settings are the same as CIFAR10. For ImageNet dataset, the parameter $\epsilon$ for NTK trace calculation is set to 0.001, the learning rate is initialized to 0.01 and weight decay is set to $10^{-4}$ with 300 epochs training. All experiments can be performed under 8 GeForce RTX 4090 in less than 1 day.

## B.1    DIFFERENT LEARNING RATES AND MOMENTUMS

In this section, we conduct two sets of experiments. We first vary the learning rate (lr) from 1 to $10^{-4}$ while keeping the momentum (mt) fixed at 0.9, and we also conduct experiments by fixing the learning rate at 0.1 and vary the momentum from 0.1 to 0.9. We plot the NTK (Neural Tangent Kernel) trace as a function of training epochs and we record the best test accuracy in the captions after lr or mt.

Recall that sgd with momentum has update rule: $w(t+1) = w(t) - lr * v(t+1)$, where $v(t+1) = \nabla l(w(t)) + mt * v(t)$. When the learning rate or momentum is very small, the network will be hard to jump out of the local minima. This explains the phenomenon in Figures 10 and 11 where the NTK trace will first undergo a certain phase of decreasing and test accuracy will be smaller in the low lr or mt regime due to this optimization insufficiency.

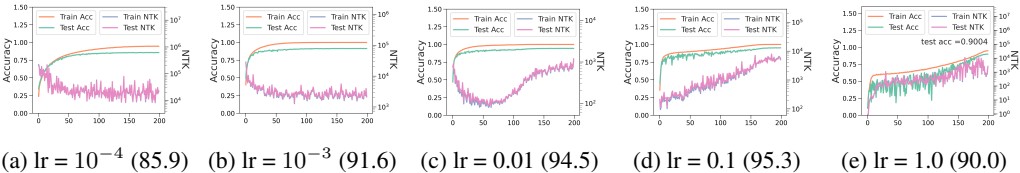

(a) lr = $10^{-4}$ (85.9)    (b) lr = $10^{-3}$ (91.6)    (c) lr = 0.01 (94.5)    (d) lr = 0.1 (95.3)    (e) lr = 1.0 (90.0)

Figure 10: NTK trace under different learning rates.

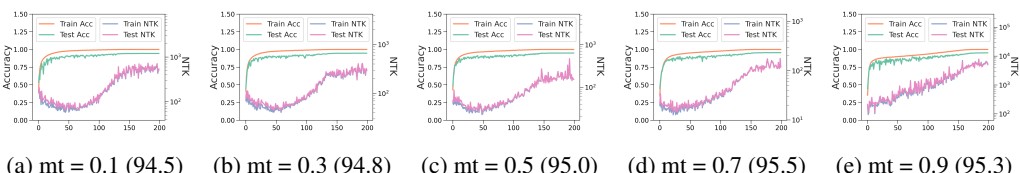

(a) mt = 0.1 (94.5)    (b) mt = 0.3 (94.8)    (c) mt = 0.5 (95.0)    (d) mt = 0.7 (95.5)    (e) mt = 0.9 (95.3)

Figure 11: NTK trace under different momentum values.

## B.2    DIFFERENT ARCHITECTURES IN A MODEL FAMILY

In the following experiment, we test the dynamic behavior of the NTK trace on a family of different models, namely ResNet18, ResNet34, ResNet50 and ResNet101. We fix the learning rate at 0.1 and momentum at 0.9, and plot the NTK trace as a function of training epochs for each model. From Figure 12, we can see the trend of NTK is similar among models.

## B.3    DIFFERENT INITIALIZATION STRATEGIES

In the following experiment, we test the influence of different initialization methods (Kaiming, Orthogonal, Xavier, Standard) on the NTK trace dynamics. We fix the learning rate at 0.1 and momentum at 0.9, and plot the NTK trace as a function of training epochs for models initialized with different methods. From Figure 13, we can see the trend of NTK is similar among different initializations.

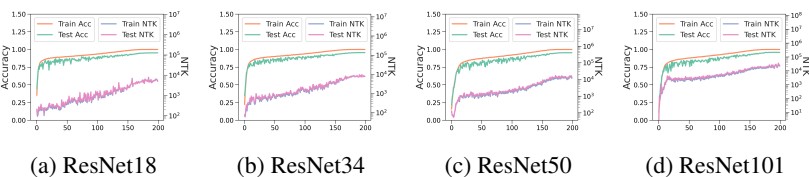

Figure 12: NTK trace of different ResNet models.

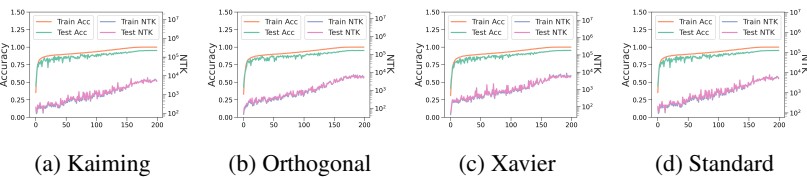

Figure 13: NTK trace under different initialization methods.

## B.4 DIFFERENT SEEDS

In this experiment, we investigate the influence of different random seeds on model training and the NTK trace. We plot both the test accuracy and train accuracy, as well as the logarithmic scale of the test NTK and train NTK trace. The left subfigures depict test accuracy and train accuracy, while the right subfigures depict the logarithmic scale of the test NTK and train NTK trace. From Figure 14, we can see the trend of NTK is similar among different seeds.

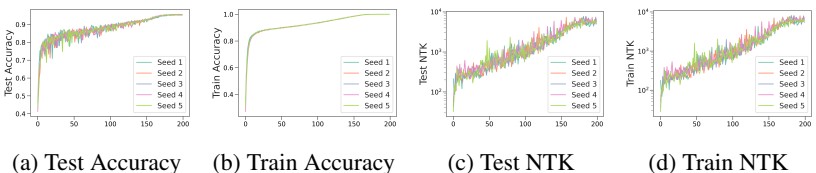

Figure 14: Effect of different seeds.

## B.5 DIFFERENT SAMPLINGS

Recall that from theorem 4.1, we have the equation $\text{Tr}(K(\mathcal{X}, \mathcal{X}; w)) = \frac{1}{\epsilon^2}\mathbb{E}_{\Delta \sim \epsilon\mathcal{N}(0,I_d)}|f(\mathcal{X}; w + \Delta) - f(\mathcal{X}; w)|_F^2$. Now, we investigate the impact of different sampled perturbations $\Delta_i$ on the approximation of the expectation. We sample 4 different perturbations and calculate the quantities $\frac{1}{K\epsilon^2}|f(\mathcal{X}; w + \Delta_i) - f(\mathcal{X}; w)|_F^2$. In Figure 15, we plot these quantities on the training and testing datasets. From the figure, we observe that different samplings have minimal effect on the pattern of NTK trace estimation. The overall trend remains consistent regardless of the specific perturbations sampled.

## B.6 DIFFERENT OPTIMIZER AND ARCHITECTURE

In our main experiments in section 4, we use the ResNet architecture and the sgd optimizer. However, to understand the behavior of the NTK trace under different optimizers or architectures, we consider the popular Adam optimizer and the ViT (base) architecture. We plot the NTK trace under these different scenarios in Figure 16. We observe that the NTK trace generally exhibits an increasing trend until it stabilizes, regardless of the optimizer or architecture used. Although the optimization speed may vary, the overall trend remains consistent. These findings provide further insights into the behavior of the network and its relationship with different optimizers and architectures.

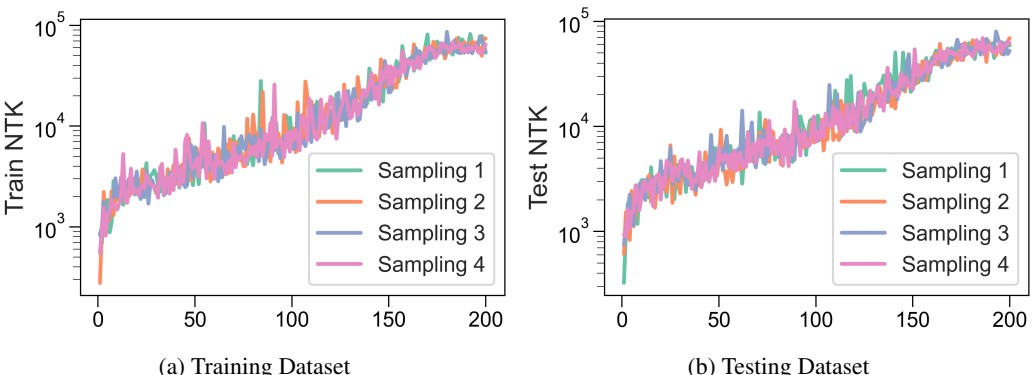

(a) Training Dataset

(b) Testing Dataset

Figure 15: The evolution of NTK trace estimations under different sampling of Gaussian perturbation of parameters, on both training and testing datasets.

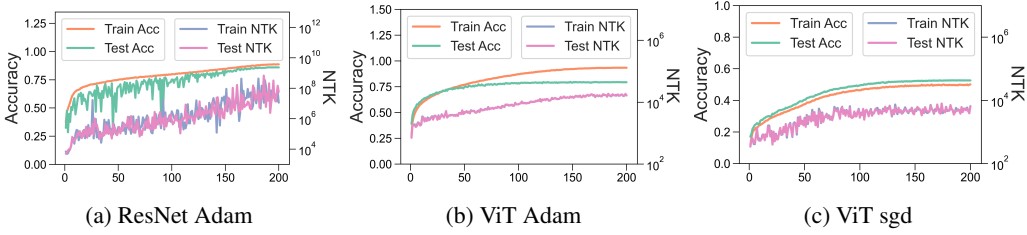

(a) ResNet Adam

(b) ViT Adam

(c) ViT sgd

Figure 16: The evolution of NTK trace estimations under different optimizers and architectures, on both training and testing datasets.