# OpenReview forum: "Understanding Neural Tangent Kernel Dynamics Through Its Trace Evolution"
_ICLR.cc/2025/Conference — Submitted to ICLR 2025_

### Official Review · Reviewer_4dGz · 2024-10-24

**Soundness:** 2
**Presentation:** 1
**Contribution:** 1
**Rating:** 3
**Confidence:** 3

**Summary:**

This paper studies the evolution of trace of the NTK matrix during training. The authors consider three settings. The first one is the supervised binary classification problem, where the authors claim that they found the connection between the NTK trace and the margin of certain kernel SVM problems. The second one is the settings where grokking happens, the authors claim that they found the NTK trace stabilizes only after the test loss stabilizes. The third one is semi-supervised learning, the authors claim that the evolution of NTK trace helps understanding the evolution of test accuracy.

**Strengths:**

1. The paper study the evolution of the trace of NTK matrix, which is an interesting topic in my opinion.

2. The experimental findings that in Figure 5,6 which connect the NTK trace to grokking seems to be interesting, assume the correctness of the experiments.

**Weaknesses:**

1. The writing of the paper needs to be improved. There are a few issues in my opinion:
-  Sometimes when citing the results from another paper, the authors do not discuss the results properly, which makes it hard to understand. To name a few:
    - In line 177, Theorem 4.2: "under some technical assumptions", please specify the assumptions, and specify the theorem from (lyu & li, 2019) that you are citing. Also, as far as I understood, a key technical assumption for the results in (lyu & li, 2019)  is that the homogeneity  of the model, I think you should at least mention this.
    - In line 192, "Motivated by (Seleznova et.al 2024), we will make the following simplification assumptions: ... ", please discuss why it is reasonable to make those simplification assumptions under the settings of Theorem 4.2. The main point I'm missing is whether the results in  (Seleznova et.al 2024) still hold under the assumptions needed for Theorem 4.2.?
    - In line 352 and the following equation (5), (6), could you specify the augmentation methods $\omega(\cdot), \Omega(\cdot).$ Also, what is the definition of the notation $prob(\cdot)$?

- Some sentences, phrases, and figures are confusing, to name a few:
    - In line 197, "Collect the dual variable in $\alpha$ which corresponds to $y_i = +1$ as $\alpha^1$ and $y_i = -1$ as $\alpha^2.$", I don't really understand this sentence.
    - The architecture of RedNet18 in Figure 3 is confusing. You call it "Layer 1,2,3,4", but as far as I understood, each of the "Layer $i$" in Figure 3 is the concatenation of a few convolution layers. I suggest either putting the true architecture figure of ResNet18 or explaining what is "Layer 1,2,3,4" in the context.
    - Line 141 and 306, the approximation of NTK trace should be $\frac{1}{\epsilon^2} \mathbb{E}\_{\Delta} [ | f(\mathcal{X}; w+\Delta) -  f(\mathcal{X}; w)  |^2 ]$, you missed $\mathbb{E}\_{\Delta}$  in these two lines.


2. The authors summarized 3 main contributions of this paper, however, each of them doesn't seem very solid to me.
- The first contribution the authors claim is "presenting an efficient for NTk trace and link it to the margin of a kernel SVM problem. "
    - First, Theorem 4,1 is wrong. Note that by assumption $\Delta \sim \epsilon \mathcal{N}(0, I_d),$ so $\mathbb{E} [ \Delta \Delta^\top ] = \mathbb{E} [ || \Delta ||_2^2] = \epsilon^2 d.$ There is a $d$ factor missing, and I'm not sure how this affects later experiment results.
    - The connection between the kernel trace and the margin of a kernel SVM problem is based on the simplification assumptions in line 194-196. As I mentioned before, it is not clear to me why it is reasonable to make these assumptions.
    - Theorem 4.3 seems to be a straightforward consequence of Cauchy-Schwarz inequality, I don't see the point to state it as a theorem.

- The second contribution the authors claim, as far as I understood, is that in some settings (supervised classification ), the NTK trace stabilizes when the training curve reaches the maximal; in grokking settings, the NTK trace stabilizes after the training process stabilizes after the testing loss reaches the maximal.
    - For the claims for grokking settings, in Line 311, you claim " Our analysis reveals that both NTK traces on both the train and test datasets stabilize only when the test accuracy approaches it maximum value." I failed to see this in Figure 5. In Figure 5, the NTK trace should only stabilize after around $10^4$ steps, when the test accuracy stabilizes. However, I didn't see a significant difference in the stability of the NTK trace curve between the $10^2 - 10^4$ steps and after $10^4$ steps. There's a sudden drop at $10^4$ but you only plot a very short time after $10^4$ steps, and it is not clear to me whether it reaches a plateau.
    - The same happens in Figure 1, especially in Figure 1 (a) and (c), it is not obvious that the curve of the NTK trace reaches a plateau after the training accuracy stabilizes. I think maybe you could try to smooth the curve a bit by doing a few more experiments and taking the average, etc. Also, I suggest running the experiments a bit longer after the train\test error stabilizes so that one can see a clear plateau of the curve (like you do in Figure 2b).

- The third contribution the authors claim is that the NTK trace helps to understand the training dynamics in semi-supervised learning. However, I completely failed to understand the results presented in section 5.3.
    - In Theorem 5.2, you give a lower bound on the square root of training error, and you plot the lower bound in Figure 9(b). First of all, you say the ratio is decreasing as $\tau$ is increasing, but it seems to be that in the end, the ratio will converge to zero. Also, "this matches the fact that test accuracy is close but with decreasing order with the size of $\tau$." I failed to see the connection. First of all, the ratio is a lower bound on the training loss, not the test loss; besides, it is only a lower bound and is far from tight. Could you explain more about the connection between Figure 9(b) and the test accuracy?

**Questions:**

Please also refer to the strengths and weaknesses part.

1. I'm curious about the universality of the results in Figure 1 and Figure 5,6. In the paper, you only try one experiment on three datasets for non-grokking settings and two different experiments for grokking settings. I wonder whether those experimental findings also appear in different settings, such as for different tasks, for different models, for different datasets.

---

### Official Review · Reviewer_SRnq · 2024-10-29

**Soundness:** 3
**Presentation:** 2
**Contribution:** 2
**Rating:** 3
**Confidence:** 4

**Summary:**

Through a series of experiments, the paper studies the NTK trace evolution for ResNet18 trained in various setups. The findings are the following:

1. In a standard supervised learning setting, the trace monotonically increases until the maximum training accuracy is reached; at this point, the trace stabilizes. Under some simplifying assumptions, the NTK trace is related to the margin of a classifier. Therefore maximizing the trace is equivalent to maximizing the margin.
2. In a "grokking" scenario, the trace stabilizes only when the test accuracy reaches its maximal value.
3. In a semi-supervised learning setting, the trace exhibits two phases: a decreasing phase, and an increasing one. Remarkably, the time when the phases switch does not depend on the amount of labeled samples.

The motivation behind studying the NTK trace is that it is an integral scalar characteristic of the NTK that can be efficiently approximated.

**Strengths:**

**Originality**

I haven't seen the idea of studying the trace of the NTK as its integral characteristic in the literature before, as well as the approximation formula in Th.4.1.

**Significance**

The paper studies diverse scenarios: supervised learning, grokking, and semi-supervised learning.

**Weaknesses:**

* In my opinion, the main weakness of the paper is the lack of solid insights. In short, what they observe is that the NTK trace monotonically increases in supervised learning (including grokking), and first decreases and then increases in semi-supervised learning. The authors do not provide the reader with any concrete conclusions from these observations. To be specific, looking at the contributions section (lines 60-69), we see quite generic statements; they are more detailed in Section 6, Conclusions, but still, no concrete insights are provided.
* To link the NTK trace to the margin, they assumed that according to the "after-kernel" (NTK at the end of training), all examples of the same class are equally similar, while all examples of different classes are orthogonal (lines 192-197). These assumptions seem too strong to me; some experimental evidence supporting these assumptions would be nice.
* According to lines 207-208, the purpose of Figure 1 is to demonstrate that "the network is implicitly maximizing the margin of kernel SVM during training". However, this statement has already been proven theoretically in [1], Corollary 4.5.
* The paper has conflicting notation: K for the kernel, and K for the output size. This is specifically confusing in line 132. Also, x for an input, and x for a kernel diagonal value.
* It should be specified that $\sim$ in Theorem 4.1 means "equivalent as $\epsilon \to 0$".
* I would prefer to have an explicit mapping between the theorems in the main and those in Appendix A.

**Questions:**

Could the authors comment on why the NTK trace is an important object to study, and what kind of insights the evolution of trace provides?

---

### Official Review · Reviewer_tsi9 · 2024-10-31

**Soundness:** 2
**Presentation:** 3
**Contribution:** 1
**Rating:** 3
**Confidence:** 4

**Summary:**

This paper investigated the behavior of NTK by examining its trace during training. As claimed, the NTK trace typically exhibits an increasing trend and stabilizes when the network achieves its highest accuracy on the training data. Moreover, it involves an intriguing scenario where the test accuracy suddenly improves long after the training accuracy plateaus.

**Strengths:**

This paper is easy to follow.

**Weaknesses:**

There is a conflict between the idea NTK and the empirical one; the former is derived by gradients and initialized parameters that obey the Gaussian distribution, whereas the latter relates closely to the trained parameters. According to the NTK derivation, the empirical one cannot follow the Gaussian distribution, but resulting in an empirical kernel. Thus, the results of Theorem 4.1-4.2-4.3 do not hold for the standard NTKs. It just likes an optimization problem using deep neural networks. Therefore, the significance of investigating the convergence of this empirical kernel is limited.

**Questions:**

nothing.

---

### Official Review · Reviewer_z5Gj · 2024-11-03

**Soundness:** 3
**Presentation:** 3
**Contribution:** 3
**Rating:** 6
**Confidence:** 3

**Summary:**

The paper explores the Neural Tangent Kernel (NTK) in the finite-width regime by analyzing the evolution of its trace during neural network training. It covers three experimental settings: supervised classification, the grokking phenomenon, and semi-supervised learning. The authors propose an efficient approximation for the NTK trace and provide empirical evidence linking the NTK trace dynamics to model performance. The paper aims to improve understanding of NTK's behaviour in modern neural networks and demonstrates its relevance in various training scenarios.

**Strengths:**

* The authors provide a thorough exploration of NTK trace dynamics across different training settings, offering new insights into neural network behaviour.

* The introduction of a computationally efficient method for approximating the NTK trace is a strong contribution, making the analysis more feasible for large-scale models.

* The theoretical connection between the NTK trace and the margin in an SVM problem adds a robust mathematical foundation to the findings.

**Weaknesses:**

* While the paper focuses on the NTK trace, it does not thoroughly explore other essential properties of NTK that might impact training dynamics.

* Some of the equations and theoretical explanations are challenging to follow due to inconsistencies in notation and unclear terminology, e.g., The transition from vector notation ∥⋅∥F (Frobenius norm) to the trace notation Tr(⋅) in Theorem 4.1 requires a clearer explanation. variables like w are used for neural network parameters and the optimal margin vector in the SVM context. This dual use of w could confuse readers.

* The empirical observations are well presented, but the analysis could be deeper. The implications of NTK trace behaviour, especially in semi-supervised learning and grokking, require more detailed discussion.

**Questions:**

1. It remains unclear how the insights gained from NTK trace dynamics can be directly applied to improve neural network training or architecture design. A discussion on potential applications or improvements is necessary.

2. Terms like "grokking" are introduced but could benefit from a more formal definition or context. While the term is defined as a phenomenon where test accuracy improves long after training accuracy plateaus, a brief explanation of why this occurs or how it differs from typical training dynamics would help readers unfamiliar with the concept.

3. The phrase "logit perturb distance" is used in the section on grokking, but it is not immediately clear how this metric is calculated or what its significance is. Providing a concise explanation or a simple illustrative example would make the concept more accessible.

---

### Meta-Review · Area_Chair_3vDu · 2024-12-06

**Metareview:**

The paper analyzes the trace of the NTK during neural network training and provides results in 3 settings: supervised learning, grokking and semi-supervised learning. Some reviewers noticed that studying (and approximating) the trace evolution is an interesting topic, and I agree with that. However, several serious issues have been raised (lack of solid insights, problems with both theoretical and experimental contributions) and the authors have not provided a response. Thus, the paper should be rejected.

**Additional Comments On Reviewer Discussion:**

Unfortunately, the authors did not respond to the comments of the reviewers, so there was no discussion.

---

### Decision · Program_Chairs · 2025-01-22

Reject